# Diagnostic Value of C-Reactive Protein and Serum White Blood Cell Count during Septic Two-Stage Revision of Total Knee Arthroplasties

**DOI:** 10.3390/antibiotics12010014

**Published:** 2022-12-22

**Authors:** Sebastian Benda, Moritz Mederake, Philipp Schuster, Bernd Fink

**Affiliations:** 1Department of Arthroplasty and Revision Arthroplasty, Orthopaedic Clinic Markgröningen GmbH, Kurt-Lindemann-Weg 10, 71706 Markgröningen, Germany; 2Department of Orthopaedic Surgery, University Hospital Tübingen, Hoppe Seyler–Str. 3, 72076 Tübingen, Germany; 3Department of Orthopaedics and Traumatology, Paracelsus Medical Private University, Clinic Nuremberg, Prof. Ernst Nathan Straße 1, 90419 Nürnberg, Germany; 4Orthopaedic Department, University-Hospital Hamburg-Eppendorf, Martinistrasse 52, 20251 Hamburg, Germany

**Keywords:** bone and joint infections, CRP, knee arthroplasty, infection parameters, orthopedic infections, periprosthetic joint infection, two-stage revision, white blood cell count

## Abstract

Aims and Methods: In septic two-stage revision arthroplasty, the timing of reimplantation is crucial for therapeutic success. Recent studies have shown that singular values of C-reactive protein (CRP) and white blood cell count (WBC count) display weak diagnostic value in indicating whether periprosthetic joint infection (PJI) is controlled or not during two-stage revision surgery of knee arthroplasty. Therefore, in addition to the values of CRP and WBC, the course of CRP and WBC counts were compared between groups with and without later reinfection in 95 patients with two-stage revision (TSR) of infected total knee arthroplasties (TKA). Of these patients, 16 had a reinfection (16.84%). Results: CRP values decreased significantly after the first stage of TSR in both the reinfection and no-reinfection groups. WBC count values decreased significantly in the no-reinfection group. Decrease in WBC count was not significant in the reinfection group. No significant difference could be found in either the CRP values or the WBC counts at the first stage of TSR, the second stage of TSR, or their difference between stages when comparing groups with and without reinfection. Area under the curve (AUC) values ranging between 0.631 and 0.435 showed poor diagnostic value for the calculated parameters. The courses of CRP over 14 days after the first stage of both groups were similar with near identical AUC. Conclusions: CRP and WBC count as well as their course over 14 days postoperatively are not suitable for defining whether a PJI of the knee is under control or not.

## 1. Introduction

Periprosthetic joint infection (PJI) is one of the most feared complications of total knee arthroplasty (TKA). The incidence of PJI following arthroplasty of the knee ranges between 1 and 2.4% [1,2,3]. Published guidelines differentiate between early (acute) and late (chronic) PJIs [4,5,6,7]. An early PJI is usually classified as diagnosed within less than 4 weeks after surgery. When diagnosed after 4 weeks, the PJI can be classified as a late infection [3,4,7,8,9].

While early or acute infections can be approached by debridement, antibiotics, and implant retention (DAIR), exchange of the prosthesis is usually necessary for successful treatment of late PJI [10]. An exchange procedure is either performed as a one-stage or two-stage revision (TSR) [3,7]. Similar good results have been reported for one-stage revisions (when the causative microorganism is known) and two-stage revisions [11,12]. However, two-stage revision is still the currently preferred therapeutic option for the treatment of late PJI. Hereby, the first surgery (first stage of TSR) involves the removal of all foreign material as well as aggressive debridement, followed by the implantation of a spacer, usually using antibiotic-impregnated polymethylmethacrylate (PMMA) cement. After surgery, patients undergo several weeks of intravenously and subsequent orally administered antibiotic treatment in accordance with the established bacterium’s susceptibility profile. In a second surgery (second stage of TSR), the spacer is removed, another radical debridement is performed, and a new prosthesis is implanted. This strategy yields success rates of 91 to 96% [3,7,8,13]. The interim phase usually takes 6 to 12 weeks. However, the duration of the interim phase can differ, and to determine the optimal time for reimplantation, most authors propose usage of serum parameters to decide if an infection is still active or not [14].

Investigating whether laboratory markers before the reimplantation of a new prosthesis are suitable for determining if an infection is still active or not, as well as setting a threshold value, has been the object of several recent studies. Parameters such as C-reactive protein (CRP), white blood cell (WBC) count, interleukin-6 (IL-6), erythrocyte sedimentation rate (ESR), and fibrinogen levels were tested as singular parameters before reimplantation in collectives consisting of TKA or mixed collectives of TKA and total hip arthroplasty (THA). However, none of these studies were able to determine threshold values with high enough sensitivity and specificity or good diagnostic value [15,16,17,18].

Therefore, the current study was conducted to determine whether the course of CRP and WBC count over several days between stages one and two of TKA yields a better diagnostic value than singular values of the same inflammatory parameters. Moreover, in most previous studies, there was no differentiation between PJI of the knee and PJI of the hip. We believe that a joint-specific diagnostic is necessary, because studies showed significantly different thresholds for CRP in PJI of TKA and THA (higher for TKA) [19]. To our knowledge, there are no specific studies addressing two-week CRP or WBC count progression in patients with PJI of the knee.

Therefore, the goal of this study was to answer the following questions:Do CRP and WBC count show a significant response to the first stage surgery of TSR and subsequent antibiotic treatment?Do CRP and WBC counts behave differently in cases of later reinfection?What is the diagnostic value of CRP and WBC counts, as well as their course over 14 days in the interim phase for predicting later re-infection?Is there a threshold value of CRP and WBC counts, as well as for their course over 14 days, with a diagnostic value that would help decision-making for reimplantation?

## 2. Material and Methods

### 2.1. Patients

The current retrospective study was approved by the local ethics board (registration number 418/2021BO2) and was performed in line with the guidelines of the Declaration of Helsinki. A prospectively collected database of patients who underwent a two-stage septic revision arthroplasty with PJI of the knee between 2013 and 2019 was used. The exclusion criteria were the following: additional inflammatory disease (rheumatic disorders, co-occurring infection of other origin such as pneumonia, compartment syndrome, multi-organ failure), incomplete data, less than 24 months of follow-up data. After applying exclusion criteria, our database consisted of 95 patients.

Data collected from patients’ medical records included age, sex, time period between explantation and reimplantation of prostheses, previous PJI, body mass index (BMI), comorbidities, the type of PMMA cement used in revision surgery, antibiotics used to treat PJI intravenously as well as orally, cell count in histopathologic samples obtained during revision surgery, causative organisms cultured from samples obtained during revision surgery, and laboratory parameters (including CRP and WBC counts, creatinine) throughout the inpatient treatment (14 days) after first-stage surgery, as well as immediately before the second stage of TSR.

### 2.2. Treatment Protocol

Prior to revision surgery, patients underwent aspiration and/or biopsy examination of the knee to confirm PJI. Aspiration was performed without anesthesia. The harvested fluid was immediately introduced into pediatric blood culture bottles containing BD BACTEC-PEDS-PLUS/F-Medium (Becton Dickinson, Heidelberg, Germany) and incubated for 14 days. If aspiration samples did not show any microorganism growth, but clinical signs of PJI still remained, patients underwent surgical biopsy with general anesthetic. The biopsy samples were each placed in sterile tubes and transferred together with the aspirated fluid to the microbiological laboratory (certified according to DIN EN ISO 15189 and DIN EN ISO/IEC 17025) within an hour of sampling. Patient specimens were processed immediately after arrival at the laboratory. PEDS culture vials were treated with Fastidious Organism Supplement (FOS) (Becton Dickinson, Heidelberg, Germany) and incubated using the BD BACTEC 9050 automatic blood culture system (Becton Dickinson, Heidelberg, Germany). Turbid broths were subcultured onto appropriate agar plates. Microorganisms were identified by standard microbiological procedures including biochemical characterization with the API system (BioMerieux, Nuertingen, Germany) in case of anaerobic bacteria. Antibiotic susceptibility testing was performed by disk diffusion or dilution methods according to the Clinical and Laboratory Standards Institute (CLSI) guidelines. In all other cases, we used Vitek II (BioMerieux, Nuertingen, Germany) for identification and antibiotic susceptibility testing. All the samples were incubated for 14 days. The results together with results of the aspiration were analyzed according to the ICM-criteria 2018 [20,21]. According to the microorganism’s antibiotic susceptibility profile, an antibiotic treatment was scheduled by the corresponding microbiologist (who is specialized in PJI). After confirmation of PJI, patients underwent a two-stage revision.

#### 2.2.1. First Stage of TSR

The first stage of TSR consisted of explantation of the infected prosthesis, obtaining histological and bacteriological samples, radical debridement, and thorough lavage and insertion of spacer components together with antibiotic-impregnated PMMA cement.

#### 2.2.2. Interval

Antibiotics were selected based on the above-mentioned culture results or empirical use in cases of preoperative negative cultures. Immediately after surgery, bacteriological and histological assessments were repeated using the intraoperative samples, so that changes in causative microorganisms could be addressed by correcting the individual specific antibiotic treatment. Each patients’ antibiotic treatment was administered individually by a microbiologist and consisted of 2 weeks of intravenously administered antibiotics, followed by another 4 weeks of oral antibiotic therapy. During the 2 weeks of inpatient treatment, serum CRP and WBC count tests were performed regularly. After 6 weeks of antibiotic treatment, the second stage of TSR surgery was performed. Prior to the second stage of TSR surgery, serum CRP values and WBC counts were obtained to assess infection status.

#### 2.2.3. Second Stage of TSR

The second stage of TSR surgery consisted of the explantation of spacer components and PMMA cement, debridement, lavage, and reimplantation of a new hinged prosthesis with individual and specific antibiotic-impregnated PMMA cement as described above. Antibiotic treatment after reimplantation followed the same regime as that after the first stage of TSR surgery.

### 2.3. Laboratory Parameters

WBC count (/μL) was measured with a fully automated hematology analyzer (UniCel DxH 800; Beckman Coulter, Pasadena, CA, USA), which identifies cells based on the principle of impedance technology and light scatter. CRP (mg/L) was measured by a particle-enhanced turbidimetric immunoassay (Cobas C303; Roche, Basel, Switzerland). Both methods were performed according to the manufacturers’ recommendations. CRP (mg/L) and WBC count (/µL) were recorded prior to the first and second stage of TSR up to 3 days before surgery and on the first postoperative day, as well as on a regular basis during the inpatient stay depending on the day of the week when the surgery was performed; other clinical symptoms were also recorded, which led to further laboratory measurements. There were on average 7.5 measurements for each patient available. Changes in those inflammatory markers were named as “ΔCRP” and “ΔWBC count” and calculated using values prior to the first stage of TSR minus values prior to the second stage of TSR surgery.

### 2.4. Outcome Measurements

Follow-up examinations took place regularly after the second stage of TSR. Treatment failure was classified as having another PJI of the knee within 24 months after the second stage of TSR. Patients were classified as free from reinfection according to Diaz-Ledezma et al. [22] if they met the following criteria: free from mortality related to PJI, free from subsequent surgical intervention for PJI, microbiological as well as clinical absence of the infection for at least 24 months and CRP <10 mg/L [3,7,20]. Values below the in-house threshold of detection were displayed as <5 mg/L. For statistical analysis to be possible, cases with CRP values <5 mg/L were virtually set to 0 mg/L. The follow-up period was 63.11 ± 19.77 months (mean ± standard deviation).

### 2.5. Statistical Analysis

IBM SPSS Version 24 (IBM Corp., Aemonk, NY, USA) and Microsoft Excel (Microsoft, Redmond, WA, USA) were used for statistical analysis. Categorical variables are depicted as frequencies, while continuous variables are shown as medians and ranges. Mann–Whitney U-tests and Wilcoxon tests were performed for comparisons, while *p*-values were calculated with an alpha-level of 0.05 (without having been adjusted for multiple testing) and are two-sided. The effect size of statistical tests was determined by calculating r and defining values of r < 0.3 as weak, r = 0.3–0.5 as moderate, and r > 0.5 as strong. To determine the diagnostic value of the diagnostic tests, receiver operating characteristic (ROC) curves were calculated. The area under the curve (AUC) was calculated for measuring diagnostic effectiveness. AUC < 0.6 was defined as diagnostic failure, AUC = 0.6–0.69 was defined as poor, AUC = 0.7–0.79 was defined as fair, AUC = 0.8–0.89 was defined as good, and AUC = 0.9–1 as excellent [23]. To define threshold values of laboratory tests, Youden’s J-statistics were performed.

## 3. Results

### 3.1. Collective

A total of 95 patients was included, out of which 79 stayed free of reinfection and 16 had a reinfection (16.8%). There were 51 infected bicondylar knee prostheses treated with articulating spacers and 44 infected hinged knee prostheses treated with static spacers.

The collective consisted of 52 male and 43 female patients, the median age was 69.78 years (median age in males: 68.92, median age in females 70.83). Diabetes was found in 23 patients (24.2%). The most detected causative bacterium was *Staphylococcus epidermidis* (20%), followed by *Staphylococcus aureus* (17.9%), and *Enterococcus faecalis* (5.3%) (see Table 1 for a complete list). Two or more causative microorganisms were found in 10 patients (10.5%). In 32 patients (33.7%), no causative bacterium could be cultivated from preoperative or intraoperative samples, while 30 patients (31.6%) had undergone previous septic revision of the knee. In 9 cases, patients underwent spacer revision before the second stage of TSR was carried out.

### 3.2. Treatment Response

In both groups (with and without reinfection), CRP values decreased significantly over the course of 14 days of intravenous antibiotic therapy from 79.5 ± 97.4 mg/L preoperatively for the no-reinfection group and 93.2 ± 71.9 mg/L pre-operatively for the reinfection group to 14.3 ± 15.6 mg/L for the no-reinfection group and 17.9 ± 11.4 mg/L for the reinfection group before reimplantation in the second stage of TSR (Figure 1).

WBC count values also decreased over the course of 14 days from 8.1 ± 3.1 × 10^3^/µL preoperatively for the no-reinfection group and 7.8 ± 2.6 × 10^3^/µL for the reinfection group to 6.2 ± 1.9 × 10^3^/µL for the no-reinfection group and 7.0 ± 1.9 × 10^3^/µL for the reinfection group before reimplantation in the second stage of TSR. However, in the reinfection group, WBC count did not decrease significantly (Figure 2).

When analyzing delta CRP and delta WBC count, no significant difference between the reinfection and no reinfection groups could be found. Since there was quite a large difference in group sizes (79 vs. 16), the effect size r of the calculation of significance of decrease of values between the first and second stage of TSR was calculated. For both delta CRP and delta WBC count, effect sizes were small at r = 0.08 and r = 0.04, respectively.

### 3.3. Diagnostic Value and Determination of Optimal Threshold

When comparing the reinfection and no-reinfection groups, no significant difference could be found for CRP and WBC count before reimplantation (second stage of TSR). There was also no significant difference in delta CRP and delta WBC count.

ROC curves were used to assess the diagnostic value of each marker. The AUC for CRP prior to the second stage of TSR surgery was 0.631, while the AUC for delta CRP was 0.435 (Figure 3, Table 2). As for WBC count, values for WBC count prior to the second stage of TSR and delta WBC count were AUC = 0.585 and 0.527, respectively (Figure 4, Table 2).

Yourden’s J analyses were performed to calculate optimal threshold values. However, none showed adequate sensitivity or specificity (Table 3).

### 3.4. CRP Value Course in the Interim Phase

The difference in CRP levels of both groups was not statistically significant at any point during the 14 days of interval (Table 4). Areas under the curve matched closely, with 1.021 for the no-reinfection group and 1.007 for the reinfection group, respectively (Figure 5).

## 4. Discussion

Commonly used parameters to determine infection status before the second stage of TSR are serum CRP and WBC counts. The singular value of CRP before reimplantation as a marker for infection control in PJI has been questioned in several studies with varying results [17,24]. Previous studies have only analyzed individual singular CRP and WBC count values in two-stage revisions (TSR) of TKA [15,16,17,18]. We therefore analyzed CRP and WBC count progress over the course of 14 days after the first stage of TSR, in addition to singular values, to see if the count progress is a better indicator for determining infection status than the singular value before reimplantation alone.

The sensitivity of 0.75 and specificity of 0.544 for CRP at a threshold of 11.35 mg/L and the calculated AUC of 0.631 prior second stage of TSR (before reimplantation) were quite low in the current study. This is in accordance with a meta-analysis of 24 studies comparing several diagnostic tests for determining the time of reimplantation in two-stage revision of TKA and THA with a similarly low sensitivity of 0.45 and specificity of 0.73 for the CRP [25]. Only Hoell et al. [26] described an excellent specificity of 0.92, as well as an AUC of 0.704 for singular CRP level with a threshold value of 25 mg/L, while Kusuma et al. [17] reported a sensitivity of 0.94 at a threshold of 177.5 mg/L. On the other hand, both studies had low sensitivities of 0.44 and 0.13, respectively. Therefore, the better specificity using higher thresholds led to lower sensitivities in these studies, and the diagnostic value for CRP as a predicting factor was still low. Moreover, the high range of thresholds for the CRP level in the different studies from the literature between 10 mg/L and 177.5 mg/L might be another sign of the weakness of this diagnostic parameter for predicting the time and possibility of reimplantation.

In addition, the delta of the CRP level between the first and second stage of TSR showed a low diagnostic value in the current study with a sensitivity of 0.188, a specificity of 0.949 and an AUC of 0.435. Khury et al. reported an AUC for delta CRP count between the two stages of 0.654 for TKA and THA [27]. The diagnostic values of WBC and delta WBC were similarly weak, with an AUC of 0.585 for the WBC before the second stage of TSR surgery and 0.527 for the delta WBC count. A similar AUC of 0.573 was seen for the delta WBC in Khury et al. while studying the two-stage revision of TKA and THA [27].

Therefore, we came to the same conclusion of weak diagnostic value for singular values of CRP and WBC as previous studies, with mostly mixed collectives of two-stage revision of TKA and THA [16,17,18,24,25,27]. Moreover, we tried to define suitable thresholds for these serum parameters by performing Yourden’s J-statistics on our set of data but failed to generate such with sufficient sensitivity or specificity. Shukla et al. came to similar conclusions when examining PJI of the hip [28]. This further confirms that CRP and WBC count are not suitable to be used as the only predictive parameters for the risk of re-infection and, therefore, in making decisions about when to perform the second stage.

The course of the CRP and WBC values as a previously unstudied parameter in the two-stage revision of TKA also failed to show any diagnostic value in the current study with regard to making decisions about when to perform reimplantation. For two-stage revisions of THA, there are two somewhat similar studies in the literature concerning the course of CRP. Li et al. [29] analyzed 74 two-stage revisions of THA with seven reinfections and found higher median pre-reimplantation CRP levels for the reinfection group. Moreover, they were able to predict a higher failure rate for cases in which CRP never dropped below 10 mg/L. On the other hand, Mederake et al. [30] did not find different courses of CRP and WBC between 106 non-reinfected and nine reinfected two-stage revisions of THA. In accordance with the latter paper, we were also unable to recreate the results of Li et al. [29] for two-stage revisions of TKA and saw that neither singular CRP or WBC count values, nor CRP- or WBC-count courses are suitable for predicting treatment success.

Other serum parameters have also been investigated in the literature: inflammatory markers such as ESR, IL-6, and fibrinogen levels in serum also seem to be ineligible for predicting treatment success [16,17,18,25]. One reason for this could be that all inflammatory markers are also influenced by other comorbidities such as inflammatory diseases, trauma, obesity, and smoking [31,32,33,34,35].

Besides serum markers, there is also the possibility of using the WBC count in synovial fluid, for which there are varying results in the literature. Some authors found that WBC in synovial fluid can accurately detect persistent infection [28,36,37], while others reported negative results [17,25]. Bian et al. calculated, in a meta-analysis of 24 studies, a sensitivity of 0.52 and a specificity of 0.66 for the WBC in the synovial fluid [25]. The cultivation of the synovial fluid in this study showed a calculated high specificity of 0.97, but an unacceptable low sensitivity of 0.18 [25].

One strength of the current study is the consistent therapeutic and operative regime, resulting in excellent reproducibility. To remove other influences on CRP values and WBC count, we excluded patients suffering from chronic inflammatory diseases or acute infections (other than PJI).

The current study has limitations, such as the small size of the reinfection group compared with the total study collective, limiting the validity of the statistical tests. However, small group sizes with re-infection are a sign of treatment success and are a common problem in the literature [15,16,17,28]. A multicenter study could be carried out in the future to tackle this problem. Another possible limitation is that CRP values below the threshold of 5.0 mg/L were defined as negative and not further evaluated by the laboratory. This could have resulted in overlooking small relative changes of the values. However, other authors have investigated such relative changes and found them to be of little diagnostic value [16,18].

## 5. Conclusions

In conclusion, CRP and WBC counts before reimplantation, as well as the courses of those parameters over 14 days, are not helpful in identifying persistent infections in cases of PJI of the knee and therefore cannot be used for decision-making with regard to when to perform the reimplantation of the new knee prosthesis in the second stage. For predicting the treatment success of PJI, one cannot rely on inflammatory parameters alone. Algorithms need to be developed that take into account multiple factors such as serum and synovial parameters, as well as clinical presentation.

## Figures and Tables

**Figure 1 antibiotics-12-00014-f001:**
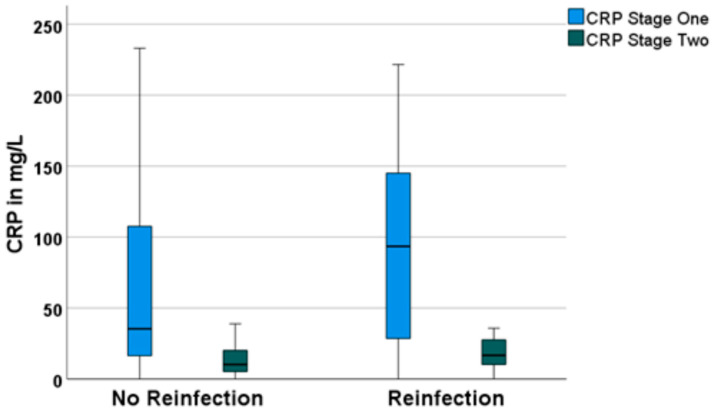
Boxplots for C-reactive protein (CRP) values of the first stage of TSR and the second stage of TSR, grouped according to “no reinfection” and “reinfection”.

**Figure 2 antibiotics-12-00014-f002:**
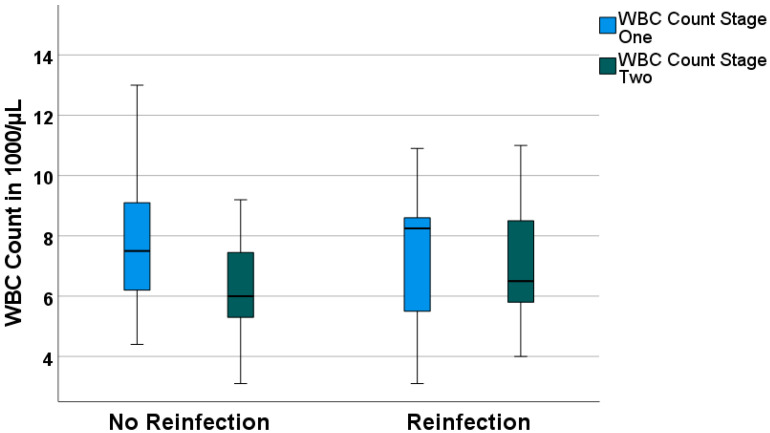
Boxplots for white blood cell count (WBC count) values of the first stage of TSR and the second stage of TSR, grouped according to “no reinfection” and “reinfection”.

**Figure 3 antibiotics-12-00014-f003:**
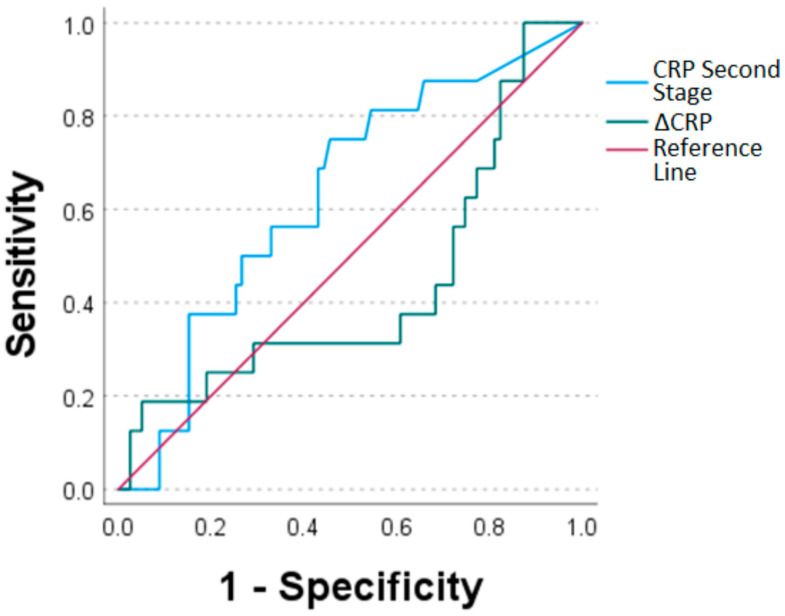
Receiver operating characteristic curves for CRP values sampled prior to the second stage of TSR (CRP second stage) and ΔCRP (CRP second stage minus CRP first stage of TSR).

**Figure 4 antibiotics-12-00014-f004:**
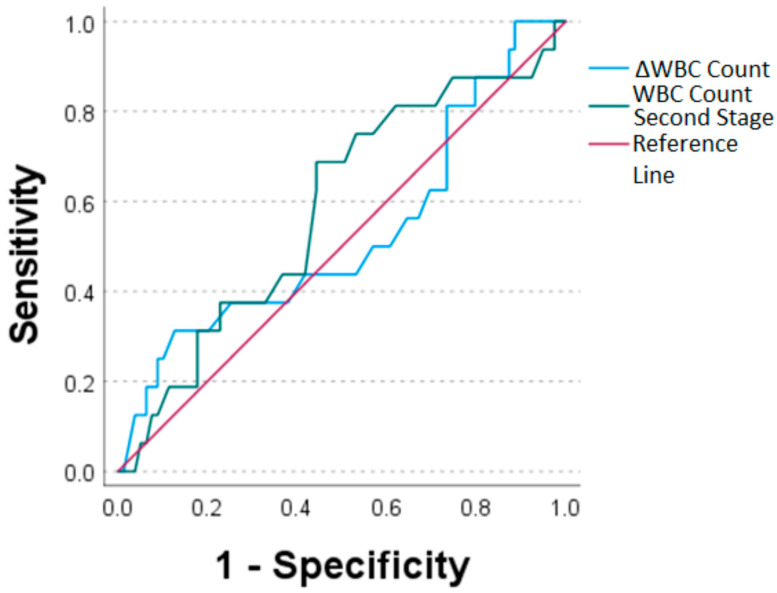
Receiver operating characteristic curves for WBC count values sampled prior to the second stage of TSR (WBC count second stage) and ΔWBC count (WBC count second stage minus WBC count first stage of TSR).

**Figure 5 antibiotics-12-00014-f005:**
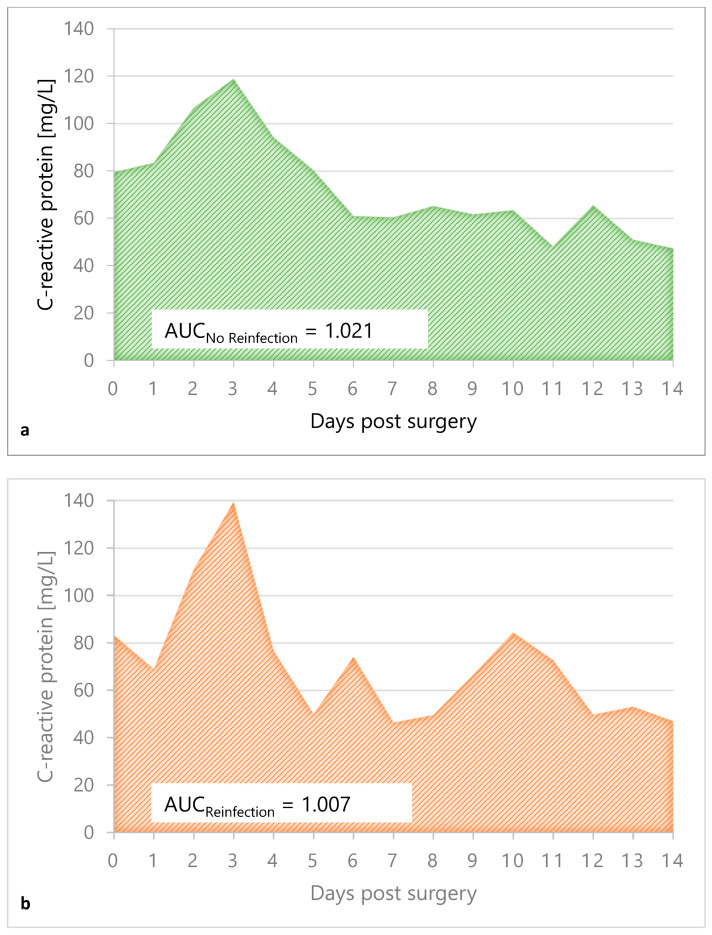
Area under the curve of mean CRP value for 14 days of follow-up after the first stage of TSR surgery of the “no reinfection” group (**a**), in green, and the “reinfection” group (**b**), in orange.

**Table 1 antibiotics-12-00014-t001:** List of all causative bacteria cultivated from preoperative or intraoperative samples and the number of patients in which each was found.

Causative Bacterium	Quantity
*Staphylococcus* *epidermidis*	19
*Staphylococcus* *aureus*	17
*Enterococcus faecalis*	5
*Staphylococcus* *capitis*	4
*Streptococcus mitis*	3
*Staphylococcus* *caprae*	2
*Streptococcus anginosus*	2
*Cutibacterium acnes*	2
*Streptococcus oralis*	2
*Pseudomonas aeruginosa*	2
*Staphylococcus hominis* ssp *hominis*	2
*Streptococcus dysgalacticae*	1
*Streptococcus parasanguis*	1
*Corynebacterium* *jeikeium*	1
*Escherichia* *coli*	1
*Staphylococcus* *agalacticae*	1
*Finegoldia magna*	1
*Streptococcus gordonii*	1
*Staphylococcus* *lugdunensis*	1
*Actinomyces odontolyticus*	1
*Staphylococcus* *warneri*	1
*Corynebacterium* species	1
*Streptococcus agalacticae*	1
culture negative	32
total	104

**Table 2 antibiotics-12-00014-t002:** The test result variables: CRP second stage of TSR, WBC count second stage of TSR, ΔWBC count have at least one tie between the positive actual state group and the negative actual stage group.

Area under the ROC Curve					
Test Result Variable (s)	Area	Std. Error	Asymptotic Sig.	Asymptotic 95% Confidence Interval	
				Lower Limit	Upper Limit
CRP count, second stage of TSR	0.631	0.073	0.072	0.488	0.774
ΔCRP	0.435	0.085	0.446	0.268	0.602
WBC count, second stage of TSR	0.585	0.077	0.270	0.434	0.737
ΔWBC count	0.527	0.086	0.750	0.360	0.695

**Table 3 antibiotics-12-00014-t003:** Calculated threshold values for C-reactive protein sampled prior to the second stage of TSR surgery (CRP at second stage of TSR), ΔCRP (CRP at second stage of TSR minus CRP at first stage of TSR), white blood cell count sampled prior to the second stage of TSR (WBC count at second stage of TSR) and ΔWBC count (WBC count at second stage of TSR minus WBC count at first stage of TSR).

	Calculated Threshold Value	Sensitivity	Specificity
CRP second stage of TSR (in mg/L)	11.35	0.75	0.544
ΔCRP (in mg/L)	11.00	0.188	0.949
WBC Count second stage of TSR (in/µL)	6.25 × 10^3^	0.688	0.557
ΔWBC Count (in/µL)	0.45 × 10^3^	0.313	0.873

**Table 4 antibiotics-12-00014-t004:** Mean CRP values with standard deviation the day before and days 1–14 after the first stage of TSR surgery (in mg/L).

	No Reinfection	Reinfection	*p*-Value
Pre-operatively	79 ± 97	83 ± 75	0.750
Day 1	83 ± 92	69 ± 58	0.879
Day 2	107 ± 80	111 ± 85	0.828
Day 3	119 ± 82	139 ± 49	0.160
Day 4	94 ± 52	77 ± 61	0.837
Day 5	80 ± 47	50 ± 42	0.207
Day 6	61 ± 30	74 ± 57	0.666
Day 7	60 ± 43	46 ± 7	0.690
Day 8	65 ± 37	50 ± 42	0.509
Day 9	62 ± 42	67 ± 80	0.622
Day 10	63 ± 45	84 ± 57	0.399
Day 11	48 ± 32	73 ± 49	0.240
Day 12	65 ± 42	50 ± 34	0.556
Day 13	51 ± 37	53 ± 50	0.768
Day 14	47 ± 38	47 ± 30	0.754

## Data Availability

Data are available from the authors on reasonable request.

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
