# Peer review of "Diagnostic Value of C-Reactive Protein and Serum White Blood Cell Count during Septic Two-Stage Revision of Total Knee Arthroplasties"

_antibiotics, 2022, doi:10.3390/antibiotics12010014_

Round 1

Reviewer 1 Report

Dear edithor, dear authors,
thank you very much for the opportunity to review this article. The topic of CRP and leukocyte count for outcome monitoring in PJI is not new. However, a novelty is that this article is exclusively related to knee arthroplasty infections and thus I think the topic is very interesting for the readership. Unfortunately, the article in its current form shows major deficiencies and should definitely be revised in terms of content but also in terms of language.

Abstract: Please check if the journal would like to have abbreviations already explained in the abstract (such as: periprosthetic joint infection = PJI).
The abstract is concise and well formulated.

Introduction
The nomenclature of two-stage, one-stage, and stage one and stage two in two-stage revision should be reconsidered. It could confuse the reader and complicates the content presentation.

Line 37: Please check if "explantation" is the correct word.

Line 37: For the purpose of completeness, DAIR should be mentioned for early infections.

Line 39: "...less aggressive…" - the one-stage revision is characterized by a much deeper debridement of the soft tissues and longer anesthesia times. Thus, in my view, this wording is not sustainable.  

Line 39: one-stage arthroplasty should be one-stage revision.

Line 40ff: The supposedly equally good results of one-stage revision are preceded by a thorough preoperative patient selection, which should at least be mentioned.  

Line 45ff: please write polymethylmethacrylate (PMMA) before cement.

Lines 62 - 70: please revise this paragraph. Convince me and the readership why it is worth to publish these results. And they are worth it! Please make it very clear what the big advantage of this study is, that only PJI was investigated of TKA and that TKA and THA are different.

Material and method
Please put this part in passive voice and avoid "we".

Line 74f: Please remove "...of the University Hospital of Tubingen..."
--> it is mentioned at line 330ff.

Line 78: Please remove "...at the Orthopädische Klinik Markgröningen..."

Please add a paragraph about microbiological sample processing. Is the laboratory affiliated with your clinic? Is there any microbiological external cooperation? Is the laboratory certified? (national accreditation organization of the Federal Republic of Germany (DAkkS) DIN EN
ISO 15,189- and DIN EN ISO/IEC 17,025-accredited microbiological laboratory ?)

Line 119: Please reconsider the nomenclature with stage one and stage two. Of course it is clear for the informed reader what you mean, but it can lead to confusion.

Results
Please use the internationally accepted nomenclature, taxonomy, spelling and convention for bacterial names. Bacteria names are written in italics. Bacterial species are written in regular letters. Please use international abbreviations like S. aureus instead of Staph. aureus and also in italics if necessary.

Table 1: Please correct according to the above. Please give bacterial names consistently either as a written out name (better) or as an abbreviation only. Please do not mix full names and abbreviations. If you want to include culture negatives, please add them to the end of the table. Propionibacterium is Cutibacterium acnes according to valid nomenclature.

Please use passive voice here as well.

Discussion
The discussion needs to be completely revised. Also linguistically. The following is an orientation example of what should be revised:

Line 240f: Unscientifically formulated.

Line 242f: References, please.

Line 236 - 247: This belongs in the introduction and would be perfect as the final sentence of the introduction.

Line 248 - 252: In my opinion, this does not belong in the discussion.

Line 253 - 266: This part repeats M&M and results. Own P-values should not be mentioned again in the discussion. Comparative values from the mentioned publications are missing. A reference to other publications is not sufficient. Please revise.

Line 262ff: These are limitations, these are already listed at the end of the discussion.

A consecutive comparison with the current literature is missing.

Conclusion
Line 316ff: Please add Conclusion as a separate paragraph according to the requirements of the journal.

References:
Please update the references. Out of 40 references, 29 are six years old and older.

Author Response

Thank you very much for reviewing our paper and the helpfull remarks. All the comments of the reviewer are addressed and the answers are in red after each comment. Moreover, the new version of the manuscript was checked by a native speaking scientist.

Reviewer 1:

Abstract: Please check if the journal would like to have abbreviations already explained in the abstract (such as: periprosthetic joint infection = PJI). Answer: The abstract has been corrected by explaining abbreviations: “…values of C-reactive protein (CRP) and white blood cell count (WBC count) display weak diagnostic value in indicating whether periprosthetic joint infection (PJI) is controlled or not … Area under the curve (AUC) values …”
The abstract is concise and well formulated. Answer: No correction needed.

Introduction
The nomenclature of two-stage, one-stage, and stage one and stage two in two-stage revision should be reconsidered. It could confuse the reader and complicates the content presentation. Answer: in the new version there is a constant nomenclature of one-stage revision and two-stage revision: Moreover, the stage one and stage two have been defined: “Hereby, the first surgery (stage one) involves … In a second surgery (stage two), the spacer is explanted, …”

Line 37: Please check if "explantation" is the correct word. Answer: see below

Line 37: For the purpose of completeness, DAIR should be mentioned for early infections.  Answer: Line 37 has been rewritten to include a mentioning of DAIR, the word “explantation” has been changed to “exchange”: “Whilst early or acute infections can be approached by debridement, antibiotics and implant retention (DAIR), exchange of the prosthesis is usually necessary for successful treatment of late PJI”

Line 39: "...less aggressive…" - the one-stage revision is characterized by a much deeper debridement of the soft tissues and longer anesthesia times. Thus, in my view, this wording is not sustainable.  Answer: the wording has been reduced to “An exchange procedure is either performed as a two-stage or, as a one-stage revision”

Line 39: one-stage arthroplasty should be one-stage revision. Answer: is corrected.

Line 40ff: The supposedly equally good results of one-stage revision are preceded by a thorough preoperative patient selection, which should at least be mentioned.  Answer: has been corrected patient selection has been mentioned

Line 45ff: please write polymethylmethacrylate (PMMA) before cement. Answer: has been corrected

Lines 62 - 70: please revise this paragraph. Convince me and the readership why it is worth to publish these results. And they are worth it! Please make it very clear what the big advantage of this study is, that only PJI was investigated of TKA and that TKA and THA are different. Answer: The aim of the study is rewritten at the end of the introduction section.

Material and method: 
Please put this part in passive voice and avoid "we". Answer: Material and methods were put in passive voice entirely

Line 74f: Please remove "...of the University Hospital of Tubingen..."
--> it is mentioned at line 330ff. Answer: it is removed

Line 78: Please remove "...at the Orthopädische Klinik Markgröningen..." Answer: it is removed

Please add a paragraph about microbiological sample processing. Is the laboratory affiliated with your clinic? Is there any microbiological external cooperation? Is the laboratory certified? (national accreditation organization of the Federal Republic of Germany (DAkkS) DIN EN
ISO 15,189- and DIN EN ISO/IEC 17,025-accredited microbiological laboratory ?) Answer: The microbiological sample processing is described in more detail. The labor is certified and there is a coresponding microbiologist (specialized in PJI).

Line 119: Please reconsider the nomenclature with stage one and stage two. Of course it is clear for the informed reader what you mean, but it can lead to confusion. Answer: The sentence was changed to “Antibiotic treatment after reimplantation followed the same regime as after stage one surgery”

Results:
Please use the internationally accepted nomenclature, taxonomy, spelling and convention for bacterial names. Bacteria names are written in italics. Bacterial species are written in regular letters. Please use international abbreviations like S. aureus instead of Staph. aureus and also in italics if necessary. Answer: All bacterial names have been put in itallics. All names in the table were written out.

Table 1: Please correct according to the above. Please give bacterial names consistently either as a written out name (better) or as an abbreviation only. Please do not mix full names and abbreviations. If you want to include culture negatives, please add them to the end of the table. Propionibacterium is Cutibacterium acnes according to valid nomenclature. Answer: corrections have been made accordingly

Please use passive voice here as well. Answer: The part was put in passive voice

Discussion:
The discussion needs to be completely revised. Also linguistically. The following is an orientation example of what should be revised: 

Answer: The discussion section is rewritten

Line 240f: Unscientifically formulated. Answer: the line has been changed to “The value of CRP as a marker for infection control in PJI has been questioned in several studies with varying results [17,26].”

Line 242f: References, please. Answer: References have been included

Line 236 - 247: This belongs in the introduction and would be perfect as the final sentence of the introduction.             Answer: This has been erased from discussion and added to the introduction

Line 248 - 252: In my opinion, this does not belong in the discussion. Answer: this part has been erased

Line 253 - 266: This part repeats M&M and results. Own P-values should not be mentioned again in the discussion. Comparative values from the mentioned publications are missing. A reference to other publications is not sufficient. Please revise. Answer: P-values were erased. Values from other publications included

Line 262ff: These are limitations, these are already listed at the end of the discussion. Answer: The sentence was erased

A consecutive comparison with the current literature is missing. Answer: This is added in the discussion.

Conclusion:  
Line 316ff: Please add Conclusion as a separate paragraph according to the requirements of the journal. Answer: A separate paragraph “Conclusion” has been added. 

References: 
Please update the references. Out of 40 references, 29 are six years old and older. Answer: The references are updated

Reviewer 2 Report

Abstract:

·         What is the background of this study. Please mention it into first line

·         Use full name of PJI, CRP and WBC.

·         Line 18: 16.8%

·         Line 19-20: “WBC count values decreased significantly in the no reinfection group, and not significantly in the no reinfection group”. What is meant by this statement? Please amend this.

·         “No significant difference could be found in neither CRP nor WBC count of stage one” Please modify this.

·         Line 22: 0.631 and 0.435

Introduction:

·         The authors need to revise introduction carefully to make it more fruitful. The introduction is too general: Report the epidemiology of TKA focus on the study area. Please show the study gaps.

·         Line 34-35: “An early or acute PJI is usually classified as diagnosed within less than 4 weeks after surgery or within less than 4 weeks after symptom onset” Please rewrite.

·         Please make a single paragraph of 2nd, 3rd and 4th paragraph in introduction.

·         Line 59: Please write down the full name of “THA” with abbreviation in first time. after this, you can use only abbreviation.

·         Line 62: White blood cell (WBC)

Material and Methods:

·         “Material and methods” are not described well. Please address all the methods and techniques used in this study.

·         Please give the description of the Hospital? What is the bed capacity? Please refer the STROBE checklist to ensure each section of the manuscript is well written (https://www.strobe-statement.org/checklists/).

·         How did you collect biopsy/aspiration sample? What was the sample management, transportation and preservation?

·         Why did not culture the samples to make confirm infection?

·         Line 101: No need to explain about results in methodology. Please rewrite your methodology step-by-step in paragraph with headings and give an appropriate reference also.

·         How did you perform CRP and CBC?

·         What is the role of creatinine in infection control as author mentioned in Line 112?

·         Line 131-133: “In-house detection threshold for CRP was set at >=5mg/l……” please give an appropriate reference.

Results:

·         Firstly, please write down the name of all bacteria into italic. Secondly, use the full name of bacteria in first i.e. Staphylococcus aureus (S. aureus) and then you can use the short name of bacteria i.e. S. aureus.

·         Please rewrite the Table 1 by following the scientific writing style.

·         Line 179: please check the spell of “c-reaktive protein”

·         Table 4: Please amend this.

Discussion:

·         Revise the discussion accordingly. What did you expect?

·         Line 242: “Previous studies have – to our best knowledge – only analyzed….”. Not clear. Please amend this.

·         Author should revise the discussion and give proper statement if their findings are not in similar with the finding of other research group’s finding.

Author Response

Abstract:

  • What is the background of this study. Please mention it into first line Answer: a background was added.
  • Use full name of PJI, CRP and WBC. Answer: The abstract has been corrected by explaining abbreviations: “…values of C-reactive protein (CRP) and white blood cell count (WBC count) display weak diagnostic value in indicating whether periprosthetic joint infection (PJI) is controlled or not … Area under the curve (AUC) values …”
  • Line 18: 16.8% Answer: has been corrected
  • Line 19-20: “WBC count values decreased significantly in the no reinfection group, and not significantly in the no reinfection group”. What is meant by this statement? Please amend this. Answer: This passage was changed: “CRP values decreased significantly in the reinfection and no reinfection group. WBC count values decreased significantly in the no reinfection group. Decrease in WBC count was not significant in the reinfection group.”
  • “No significant difference could be found in neither CRP nor WBC count of stage one” Please modify this. Answer: corrections were made accordingly.
  • Line 22: 0.631 and 0.435 Answer: has been corrected

Introduction:

  • The authors need to revise introduction carefully to make it more fruitful. The introduction is too general: Report the epidemiology of TKA focus on the study area. Please show the study gaps. Answer: The introduction is rewritten
  • Line 34-35: “An early or acute PJI is usually classified as diagnosed within less than 4 weeks after surgery or within less than 4 weeks after symptom onset” Please rewrite. Answer: Is rewritten
  • Please make a single paragraph of 2nd, 3rd and 4th paragraph in introduction. Answer: corrections were made accordingly.
  • Line 59: Please write down the full name of “THA” with abbreviation in first time. after this, you can use only abbreviation. Answer: corrections were made accordingly
  • Line 62: White blood cell (WBC) Answer: white blood cell (WBC) count was added to the list: “Parameters such as C-reactive protein (CRP), white blood cell (WBC) count, interleukin-6 (IL-6), erythrocyte sedimentation rate (ESR) and fibrinogen levels …”

Material and Methods:

  • “Material and methods” are not described well. Please address all the methods and techniques used in this study. Answer: is done
  • Please give the description of the Hospital? What is the bed capacity? Please refer the STROBE checklist to ensure each section of the manuscript is well written (https://www.strobe-statement.org/checklists/). Answer: We are an orthopaedic clinic with nearly 13000 operations per year in 251 beds. We do not think that this is relevant for the description but we would add that if the reviewer and editor want. Moreover, reviewer 1 wanted that we delete the name of our clinic.
  • How did you collect biopsy/aspiration sample? What was the sample management, transportation and preservation? Answer: Explanation of aspiration samples and biopsy have been added:

  • Why did not culture the samples to make confirm infection? Answer: The samples were cultured. This is described in more detail in the new version of the manuscript.
  • Line 101: No need to explain about results in methodology. Please rewrite your methodology step-by-step in paragraph with headings and give an appropriate reference also. Answer: methodology was devided into three paragraphs with headings
  • How did you perform CRP and WBC? Answer: This is described in more detail at the begin of chapter „Laboratory parameters“.
  • What is the role of creatinine in infection control as author mentioned in Line 112? Answer: “creatinine” was erased as it was not used to asses infection control.
  • Line 131-133: “In-house detection threshold for CRP was set at >=5mg/l……” please give an appropriate reference. Answer: There was a mistake, which is corrected and references are added.

Results:

  • Firstly, please write down the name of all bacteria into italic. Secondly, use the full name of bacteria in first i.e. Staphylococcus aureus (S. aureus) and then you can use the short name of bacteria i.e. S. aureus. Answer: All bacterial names have been put in itallics. All names in the table were written out.
  • Please rewrite the Table 1 by following the scientific writing style. Answer: corrections have been made accordingly
  • Line 179: please check the spell of “c-reaktive protein” Answer: “C-reaktive” has been changed to “C-reactive” throughout the whole document.
  • Table 4: Please amend this. Answer: corrections were made accordingly.

Discussion:

  • Revise the discussion accordingly. What did you expect? Answer: The discussion is rewritten
  • Line 242: “Previous studies have – to our best knowledge – only analyzed….”. Not clear. Please amend this. Answer: This is corrected
  • Author should revise the discussion and give proper statement if their findings are not in similar with the finding of other research group’s finding. Answer: The discussion section is rewritten

Round 2

Reviewer 1 Report

Dear Edithor, Dear Authors, 

thank you very much for the revision of your article. Your changes have noticeably improved the article. Smaller changes should be made in my opinion. 

Line 126: still have problems with the wording, this one is confusing in my opinion within the Two-Stage change from Stage one to speak. Would it be possible to call it first stage and second stage? or explantation stage and reimplantation stage (please see line 362). So that it is very clear to the reader and it is different from single stage and two stage. If you change it please change it consecutively throughout the text. 

Line 129ff: This should be in the results section. 

Table 1: Rename eighth pathogen to Cutibacterium. sixteenth pathogen please delete the dot after Staphylococcus. Twenty-second pathogen Corynebacterium in straight. 

Figure 5: Please label the X and Y axis with the correct units.

The discussion is much better done.

Line 344ff: Please clearly present the limitations separately from the strengths. Limitations should be clearly identified as such by the reader.

Please check the whole document if all numerical values carry the correct unit.

Kind regards

Author Response

REVIEWER 01

Dear Edithor, Dear Authors, 

thank you very much for the revision of your article. Your changes have noticeably improved the article. Smaller changes should be made in my opinion. 

Line 126: still have problems with the wording, this one is confusing in my opinion within the Two-Stage change from Stage one to speak. Would it be possible to call it first stage and second stage? or explantation stage and reimplantation stage (please see line 362). So that it is very clear to the reader and it is different from single stage and two stage. If you change it please change it consecutively throughout the text. ANSWER: The nomination has been changed according to the recommandation.

Line 129ff: This should be in the results section. ANSWER: It is changed according to the recommandation: Collective :A total of 95 patients was included, out of which 79 stayed free of reinfection and 16 had a reinfection (16.8%). There were 51 infected bicondylar knee prostheses treated with articulating spacers and 44 infected hinged knee prostheses treated with static spacers.

Table 1: Rename eighth pathogen to Cutibacterium. sixteenth pathogen please delete the dot after Staphylococcus. Twenty-second pathogen Corynebacterium in straight. ANSWER: is coorected

Figure 5: Please label the X and Y axis with the correct units. ANSWER: Is done

The discussion is much better done. ANSWER: No changes needed

Line 344ff: Please clearly present the limitations separately from the strengths. Limitations should be clearly identified as such by the reader. ANSWER: is done

Please check the whole document if all numerical values carry the correct unit. ANSWER: is done

Reviewer 2 Report

The authors have addressed my comments/correction and revised version looks good.

Author Response

Thank you very much for the review.

No further corrections are necessary concerning the comments of reviewer 2.

Thanks again for all the comments which improved our paper significantly.